

# Low-stress livestock handling protects cattle in a five-predator habitat

Naomi X. Louchouarn[1] and Adrian Treves[2]

[1] Nelson Insitute for Environmental Studies, University of Wisconsin-Madison, Madison, USA
[2] Nelson Institute for Environmental Science, University of Wisconsin-Madison, Madison, United States of America

## ABSTRACT

Given the ecological importance of top predators, societies are turning to non-lethal methods for coexistence. Coexistence is challenging when livestock graze within wild predator habitats. We report a randomized, controlled experiment to evaluate low-stress livestock handling (L-SLH), a form of range riding, to deter grizzly (brown) bears, gray wolves, cougars, black bears, and coyotes in Southwestern Alberta. The treatment condition was supervision by two newly hired and trained range riders and an experienced L-SLH-practicing range rider. This treatment was compared against a baseline pseudo-control condition of the experienced range rider working alone. Cattle experienced zero injuries or deaths in either condition. We infer that inexperienced range riders trained and supervised by an experienced rider did not raise or lower the risk to cattle. Also, predators did not shift to the cattle herds protected by fewer range riders. We found a correlation suggesting grizzly bears avoided herds visited more frequently by range riders practicing L-SLH. More research is required to compare different forms of range riding. However, pending experimental evaluation of other designs, we recommend use of L-SLH. We discuss the cobenefits of this husbandry method.

## INTRODUCTION

Given the important role of top predators in the function and diversity of ecosystems, societies and governments are prioritizing co-existence (*Ripple et al., 2014*). Human-induced mortality is the dominant cause of mortality for large carnivores across the world. It has resulted in ecosystem degradation (*Estes et al., 2011*), and extinctions of many populations (*Ripple et al., 2014*; *Estes et al., 2011*). Success in preserving carnivore populations depends on converting competition over land and resources from lethal to non-lethal (*Ripple et al., 2014*; *Woodroffe & Ginsberg, 1998*; *Treves et al., 2017*; *Van Eeden et al., 2018*; *Boronyak et al., 2021*). Coexistence with bears and wolves in North America is a timely challenge given the protected status of these populations in some jurisdictions and the societal support for livestock grazing on public lands. As these and other carnivores have recolonized their historic ranges they encounter infrequently supervised free-ranging livestock, leading to conflict that can either spark a renewal of eradication campaigns against top predators (*e.g.*, *Williams, 2022*) or innovative coexistence strategies.

Corresponding author
Naomi X. Louchouarn,
louchouarn@wisc.edu

In the United States, agriculture and related industries contribute 1.1 trillion US dollars to the national gross domestic product (GDP), and the highest value livestock sector is cattle production (*USDA Economic Research Service, 2021*). In Canada, cattle production alone contributes more than 5 billion Canadian dollars to Canada's GDP and tens of thousands of jobs for the province of Alberta (*Lee et al., 2017*). *Lee et al. (2017)* found that more than 60% of Alberta's beef owners claimed to have lost animals to carnivores. Methods to reduce conflict include education and attractant mitigation programs (*e.g.*, BearSafe program; *Alberta Environment and Parks, 2016*) and non-lethal and lethal predator control (*e.g.*, relocations, aversive conditioning and targeted trapping and killing; *Treves, Krofel & McManus, 2016*; *Young, Hammill & Breck, 2019*; *Khorozyan & Waltert, 2020*). However, many of the methods used to deter large carnivores have either never been experimentally tested or have limited supporting data, therefore their users assume effectiveness (*Van Eeden et al., 2018*).

Livestock owners often perceive that non-lethal methods of predator deterrence are less effective than lethal methods (*Scasta, Stam & Windh, 2017*). However, recent independent research covering various carnivore species has raised doubts about the effectiveness of killing individual predators. Although there is agreement that predation vanishes when no predators exist, the quantitative relationships between predatory threats to people or property and key environmental variables, such as domestic and wild species ecologies, remain murky. Reviews examining predator removal show that removal efforts are rarely successful, even when efforts are directed at a specific individual predator blamed for property damages (*Linnell et al., 1999*; *Odden et al., 2002*; *Treves & Naughton-Treves, 2005*; *Treves, 2009*; *Treves et al., 2019*; *Lennox et al., 2018*). Multiple studies have found that lethal control of predators often has the counter-productive effect of raising livestock losses or has no effect on losses, *e.g.*, wolves in Michigan, USA (*Santiago-Avila, Cornman & Treves, 2018*) and in France (*Grente et al., 2021*), and dingoes in Australia (*Wallach et al., 2009*). These findings have spurred many independent efforts to find other approaches. In some cases, the partnership between scientists and livestock owners can lead to the transfer and dissemination of scientifically supported innovations (*Van Eeden et al., 2018*; *Ohrens, Bonacic & Treves, 2019*; *Khorozyan & Waltert, 2021*; *Khorozyan et al., 2020*; *Radford et al., 2020*).

The effectiveness of many non-lethal methods has not been evaluated using rigorous scientific experiments (*Van Eeden et al., 2018*). One such method is range riding, *i.e.,* deploying humans using non-lethal methods of predator deterrence and livestock protection. Range riding has two primary elements: the amount of human presence and the behaviors of the humans in the field. Increased human presence among livestock is assumed to deter predators and improve response time if predators are present (*Bangs et al., 2006*). But how much human presence is effective is still unknown. Furthermore, the most effective behaviors of range riders are not well understood. Indeed range riding is not well-defined and seems to be practiced in myriad forms, each likely to have varying degrees of effectiveness (*Parks & Messmer, 2016*; *Jablonski et al., 2020*). The most commonly used forms of range riding are narrowly predator-focused where riders generally hired by government agencies focus on detection and deterrence of predators (*Parks & Messmer,*

*2016*; *Wilson, Bradley & Neudecker, 2017*). Alternatively, riders may focus more on livestock vulnerability and herding practices to foster anti-predator behavior in livestock (*Parks & Messmer, 2016*; *Bruns, Waltert & Khorozyan, 2020*). Although the latter involves deterrence of predators, search for predators is lower priority compared to the concentration of effort on livestock behavior, health, and safety. These varied practices are largely driven by anecdotal experience without the benefit of empirical data and are therefore not likely to be equally effective.

We define a particular form of range riding known as low-stress livestock handling (L-SLH) which has been developed among a relatively small group of livestock owners in the North American West (Fig. 1) (*Hibbard, 2012*). We also examine how increased presence of humans using the method with different levels of experience works to deter predators.

*Low-stress livestock handling as predator deterrent:* Bud Williams and Temple Grandin first developed L-SLH to reduce stress and improve livestock health. They combined "pressure and release" herding, a form of interacting with livestock that takes advantage of livestock prey responses, to move livestock in a way that both enhances the choices and natural behaviors of the individual animal (*Hibbard, 2012*; *Grandin, 1989*). With this technique, handlers move calmly towards livestock, coming into contact with the animal's 'pressure zone', *i.e.,* the distance at which the animal will respond to the handler's presence, and uses this contact to gently move the animal according to the livestock's instinctual responses. As ungulates generally prefer to move as a herd, handlers push individual animals towards the herd, and allow the herd to move together at a calm, yet steady pace. These techniques apparently improve livestock stress, health and yield (Fig. 1) (*Hibbard, 2012*; *Grandin, 1989*; *Barnes, 2015*). This combination of techniques creates a positive association between human actions and herding, which helps make livestock more willing to remain as a herd relative to those who are aggressively handled and therefore associate herding with stress (*Hibbard, 2012*; *Barnes, 2015*). Conventional handling generally does not consider the instinctual pressure zones of the animals, and therefore forces animals together in an uncomfortable and rapid way. This can increase stress and produce a tendency towards flightiness (*Hibbard, 2012*).

Livestock owners who have used the method in regions with high predator numbers report that their livestock behave similarly to wild ungulate herds, which may reduce vulnerability to wild predation (*Zaranek, 2016*; *Mech & Peterson, 2010*). Therefore, we hypothesize that L-SLH may deter predator attacks and reduce predation by encouraging natural herding instincts that reduce ungulate vulnerability (*Zaranek, 2016*). This method may be particularly useful on extensive public lands, where other forms of deterrence may be difficult to implement (*Eklund et al., 2017*; *Stone et al., 2017*). Though *Barnes (2015)* described a quasi-experimental evaluation of L-SLH for livestock herding, it has never, to our knowledge, been experimentally investigated as a form of predator deterrence.

Here we present the first experimentally evaluation of any form of range riding and define a few of the parameters that are important to L-SLH as a non-lethal cattle protection method. We hypothesize that range riders might deter predators from cattle by two

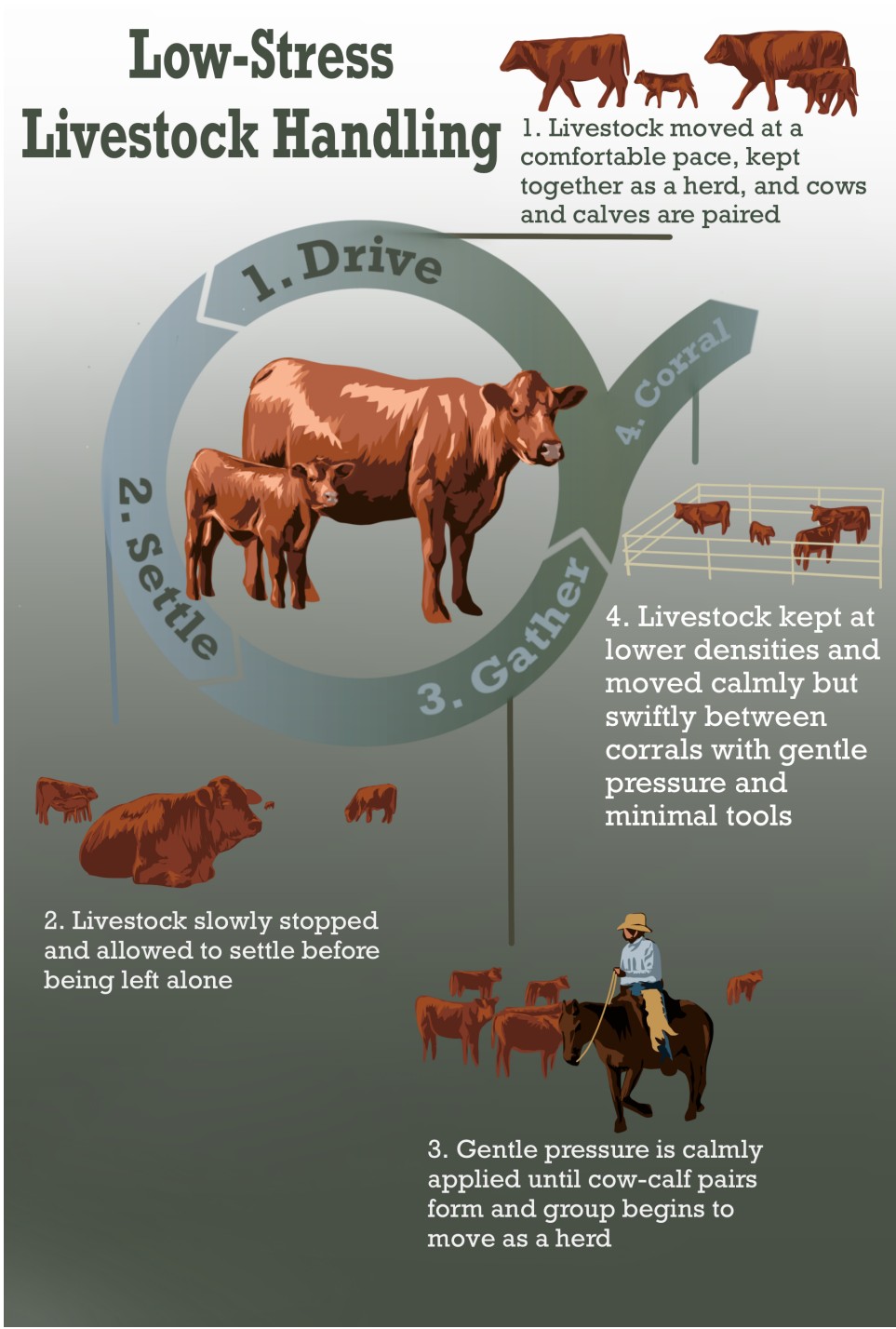

**Figure 1  A conceptual diagram of the primary elements of low-stress livestock handling.** The major elements of L-SLH are how cattle are moved (Drive), how they are stopped and placed in new pastures (Settle), how they are handled while within pastures or prepared to move out of pastures (Gather), and how cattle are corralled and prepared to be moved to new facilities or sent to feed lots (Corral).

primary mechanisms: First, the presence of more humans might deter large carnivores such as grizzly bears (*Ursus arctos*), black bears (*Ursus americanus*), wolves (*Canis lupus*), cougars (*Puma concolor*) and coyotes (*Canis latrans*; hereafter LC, for *Large Carnivores*) from the associated cattle. This predicts that the cattle guarded by a single range rider in our study (pseudo-control) would be more vulnerable than the cattle guarded by several range riders (treatment). The comparison in this study is therefore between the presence of a single rider acting as a baseline (which we define as a pseudo-control), relative to the increased human presence of multiple new riders. Alternatively, a second mechanism might be that herd stress levels predict vulnerability to predation because L-SLH would encourage and reinforce herding behaviors that reduce the risks posed by LCs. This alternative hypothesis predicts that the number of range riders is irrelevant and both pseudo-control and treatment would be effective. Also, the experience level of the range riders might affect the stress and hence vulnerability of the cattle. Accordingly, our experiment was designed to reveal if, counter-intuitively, LCs approached herds with a greater number of less experienced range riders more often, because cattle would be more stressed than cattle exposed to a single experienced range rider. Finally, a frequent unsubstantiated claim about non-lethal methods of predator control is that predators will shift to less-protected neighbors. Our design can detect this effect if herds frequently visited by multiple range riders (treatment) were visited less often by LCs than those supervised by a single range rider (pseudo-control).

## MATERIALS & METHODS

*Study area:* We conducted this study on the Spruce Ranching Co-op (hereafter the Co-op), a grazing area used by 38 permitted cattle owners who collectively bring about 2,000 cow-calf pairs and 500 pregnant heifers in June of each year (Fig. 2). The Co-op is located on 22,500 acres (91 km$^2$) of Alberta provincial lease land in the Pekisko Heritage Rangelands area, which is part of the foothills of the southern Canadian Rocky Mountains south of the Banff-Jasper-Yoho National Park complex and north of the Waterton Glacier International Peace Park (Fig. 2). The Co-op overlaps two sources of LCs, and therefore represents a core connectivity area for many species (*Proctor et al., 2012*). Despite its status as provincial lease land, we did not require permits to access the Co-op as we were not handling animals or collecting samples. Further, we used non-invasive observation methods (details below) and therefore received a study exemption from the Institutional Animal Care and Use Committee at the University of Wisconsin-Madison.

*Ranch manager:* This study required a pseudo-control, *i.e.,* a baseline condition, which we defined as the presence of the ranch manager. We use a pseudo-control instead of a placebo-control because a placebo-control requires the treatment to be compared against the lack of treatment, which in this case amounts to no human supervision. However, in this study a true control situation cannot be created because livestock owners are unwilling to leave their livestock unattended due to known risk of LC attacks on cattle. At least four individual cattle had been injured or killed by LCs in each of the past three years prior to our study. Provincial statistics on cattle predation in our study region, which is comprised

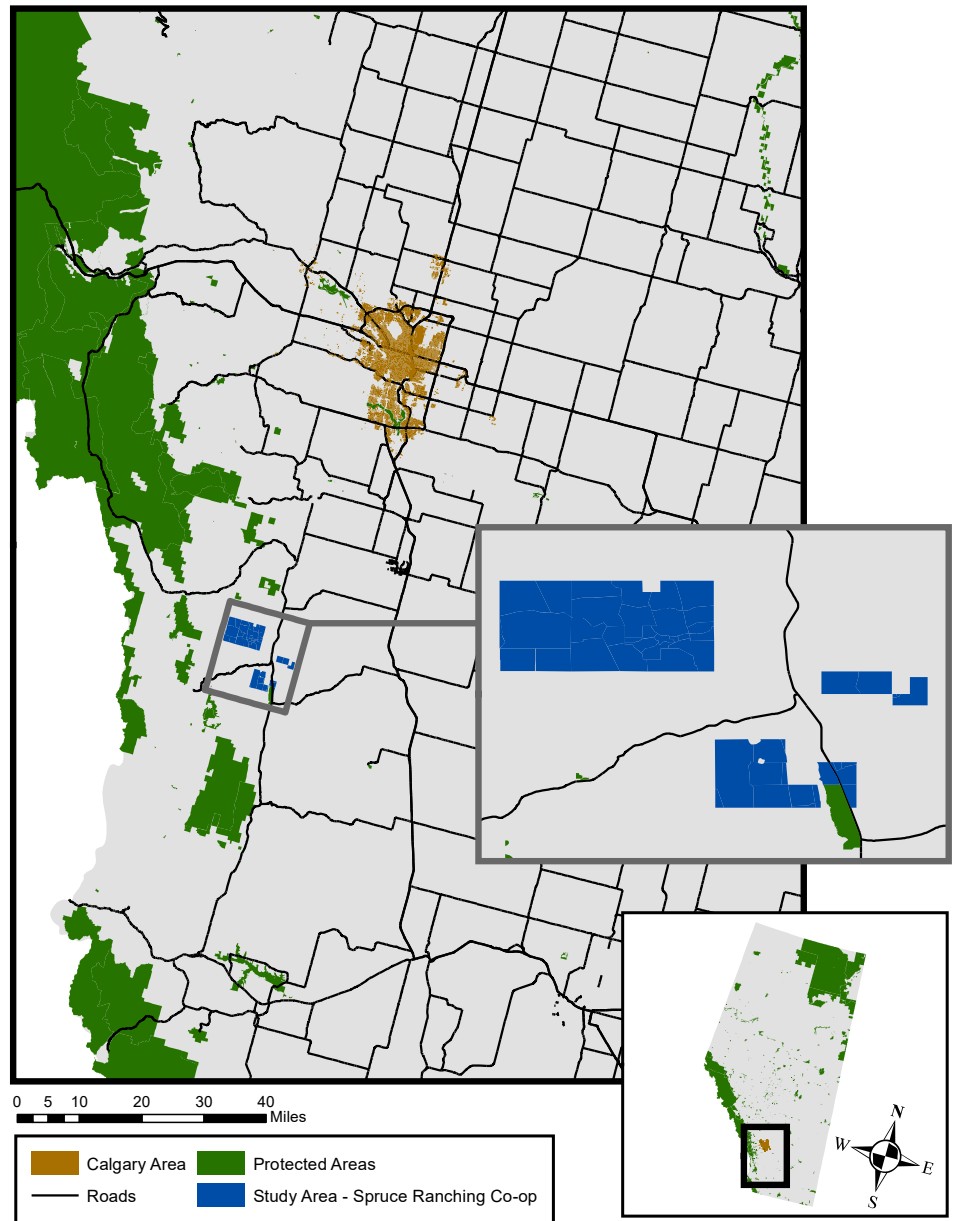

**Figure 2** **Study area map of the Spruce Ranching Co-op.** Extent map of the Spruce Ranching Co-op, located in the foothills of the Rocky Mountains in Southwestern Alberta, Canada. Protected areas refers to areas designated either by the Canadian federal government or the Alberta provincial government as protected. Base map source: Alberta Environment and Parks, Government of Alberta, 2021.

of rugged public lands where human supervision is scarce, reveal that 476 animals were confirmed attacked by carnivores between 2015–2019 (*Alberta Environment and Parks, 2020*). Also, cattle producers systematically under report suspected predation events in Alberta due to the perceived effort involved in reporting losses (*Lee et al., 2017*). These data provide some confidence that the absence of human supervision would be riskier for cattle.

The ranch manager represents a pseudo-control because: (1) The ranch manager continued the same practices he has employed for the past 20 years. The ranch manager is highly trained in L-SLH as he has practiced these behaviors on this ranch for two decades. He began learning these methods by attending clinics with Whit Hibbard (see introduction), and now travels throughout the US and Canada discussing L-SLH with interested ranchers through workshops presented by The Working Circle (https://www.workingcircle.org/), a California based non-profit. During our study, he visited every herd (*i.e.,* all eight grazing units) at the same rate he has always done without consideration of the treatment (two range riders) schedule; (2) When the ranch manager required extra help in pseudo-control fields (*i.e.,* where range riders could not go), he hired additional help, as he would normally do. The ranch manager therefore saw each herd on average every 9 days depending on the herd and changing conditions related to weather, his own schedule, and availability of hired helpers. Therefore, we compare the pseudo-control of ranch manager to the treatment of ranch manager with two recently trained range riders. We could only indirectly infer whether range riders are more effective than no protection.

*Range riders:* We hired two range riders (referred to as Rider A and Rider B). Protocols for this study did not require IRB approval as no data about range riders was collected. The ranch manager trained both range riders. Rider A has run their own ranch in the past and had attended L-SLH clinics with Whit Hibbard. Rider A was also familiar with the landscape but had never spent full seasons working with cattle on the Co-op before a short two-month pilot season in 2019. During the 2019 pilot season, the ranch manager trained Rider A for a week, during which time Rider A shadowed the ranch manager and spent supervised time moving cattle and learned to travel efficiently across the Co-op. Rider A also received a few days of training over the past two decades when the ranch manager hired them. Rider B had worked on the Co-op in the past, with about two decades of experience on their own land and on the Co-op. Therefore, Rider B already had some knowledge of how to travel safely and efficiently across the Co-op. However, Rider B had never been trained formally in L-SLH methods.

During our study, Riders A and B received a week of training from the ranch manager, during which time they shadowed the ranch manager and helped move and manage cattle. This implies some potential "blurring" of treatment and pseudo-control as the idiosyncrasies of the ranch manager's methods might have been transmitted to the two inexperienced riders. However, the ranch manager visited all herds including the treatment herds so his presence and behavior were a background baseline to the entire experiment and every herd (therefore a pseudo-control) and we were investigating the effect of the supplemental riders on half of the herds. Riders were responsible for the same herds, numbering up to 5 herds at a time. Riders worked together as a pair in areas where bears had recently (within the past 2 weeks) been seen, or that were heavily wooded where bears are difficult to detect. Riders worked alone when they deemed there to be no safety advantage from working in pairs. In other words, in fields with more open space. Riders therefore worked in pairs 38% of the time. All treatment herds received visits from riders

working alone and in pairs. Rider A helped train Rider B who was less well versed in L-SLH. Rider pairs or single riders saw each treatment herds every 1–3 days.

*Low-stress livestock handling techniques:* L-SLH practices included keeping track of cattle behaviors during range riders' visits and encouraging cattle to bunch together when they appeared to be spread too widely across a pasture. Range riders moved cattle as described on Fig. 1. For example, range riders: opened gates between pastures to allow cattle to move freely at their own pace, herded the cattle slowly to allow all cows to pair up with their calves before moving to new pastures, and used temporary pastures between original and destination pastures for up to a week to keep cattle calm throughout the movement process. Therefore, stressful movements were a minority of study period. Range riders rarely, if ever, rode their horses at more than a walk when near or among cattle. Range riders never used more assertive herding behaviors such as elevated voices or swinging arms. Range riders only used roping when doctoring animals, which was only done within a doctoring pen away from other animals. Range riders moved herds on average 3 times over the 123-day study. The ranch manager was present any time cattle were moved between pastures, but riders only participated in treatment herd moves.

When range riders found dead cattle, they contacted Alberta Fish and Game (AFG) to determine the cause of death. When AFG conservation officers implicated an LC in the death, range riders spent more time with the herd, ensuring that predators did not return. If LCs were spotted, range riders or conservation officers used aversive methods (*e.g.*, warning shots, bear bangers and cracker shells). When possible, the ranch manager moved cattle to a new field further from recent LC presence.

*Study design:* Each herd of cattle, which was independently grazed for the duration of the study period, was one 'subject' (*Ohrens, Bonacic & Treves, 2019*). Each subject herd ranged in size from 150 yearlings to 400 cow-calf pairs. Each treatment condition had an equal number of calves due to the cross-over design (Table 1). However, the average number of calves was somewhat higher in treatment condition during phase 1 (Table 1). In phase 2 that reversed and there were slightly more calves in the pseudo-control condition. By then calves would have gained weight, reducing their vulnerability, so we address this potentially confounding effect below. We conducted the experiment in two phases, each lasting half of the 4-month (July–October 2020) study period, during which we randomly assigned herds to receive 'treatment' or 'pseudo-control'. Phase 1 lasted for half of the grazing season (about 2 months; July–August) and then reversed in phase 2 (September–October) for the crossover. Therefore, treatment herds eventually became pseudo-control and vice versa. We imposed a 7-day washout period between phase 1 and phase 2 to reduce any potential carryover effect. We then compared herds to themselves from phase 1 to phase 2 to evaluate the treatment effect. Our within-subjects design minimized potentially confounding variables between herds and randomization avoided selection bias and potentially confounding order effects due to seasonal changes. We studied eight herds (replicates) and therefore had 16 trials in total, including eight treatments and eight pseudo-controls.

**Table 1  Large carnivore visits by species.** Number of large carnivore (LC) visits by subject herds, species, and phase, recorded using LC surveys and trail cameras. Number of visits are reported as visits per 10 LC surveys, visits per 100 camera-days. Treatment sequences are pseudo-control-treatment (PC-T) and treatment-pseudo-control (T-PC).

| Herd | Treatment Sequence | Number of cattle, number of calves | Wolf Visits Phase | | Bear Visits Phase | | Cougar Visits Phase | | Coyote Visits Phase | |
|---|---|---|---|---|---|---|---|---|---|---|
| | | | *1* | *2* | *1* | *2* | *1* | *2* | *1* | *2* |
| 1 | PC-T | 400, 400 | 2.0, 0 | 5.0, 0 | 8.0, 5.0 | 7.5, 1.0 | 0, 0 | 2.5, 0 | 4.0, 0 | 5.0, 2.1 |
| 2 | PC-T | 355, 355 | 0, 0 | 5.0, 0 | 6.3, 3.9 | 6.7, 1.9 | 0, 0 | 0, 0 | 2.5, 7.9 | 3.3, 0 |
| 3 | PC-T | 82, 82 | 0, 0 | 2.3, 0 | 4.4, 2.1 | 7.1, 0.8 | 1.1, 0 | 0, 0 | 4.4, 0.9 | 4.3, 2.3 |
| 4 | PC-T | 161, 0[*] | 0, 0 | 0, 0 | 2.8, 1.6 | 8.0, 0 | 0, 0 | 0, 0 | 5.7, 0.6 | 6.0, 2.4 |
| 5 | T-PC | 250, 250 | 0, 0 | 2.9, 0 | 1.4, 1.3 | 2.8, 0.63 | 0, 0 | 0, 0 | 1.4, 0 | 10.0, 0 |
| 6 | T-PC | 480, 480 | 2.5, 0 | 3.3, 0 | 11.3, 2.1 | 6.7, 4.3 | 0, 0 | 0, 2.6 | 1.3, 0.3 | 1.7, 3.4 |
| 7 | T-PC | 361, 361 | 3.8, 0.7 | 5.7, 0 | 3.7, 4.0 | 4.2, 7.5 | 0, 0 | 0, 0 | 2.5, 1.3 | 4.3, 3.0 |
| 8 | T-PC | 380, 0[*] | 2.5, 0 | 6.7, 0 | 2.5, 0 | 6.7, 0 | 0, 0 | 0, 0 | 2.5, 6.7 | 0, 0 |

**Notes.**

[*]Herds with no calves were all pregnant yearlings, *i.e.*, heifers.

The ranch manager split one herd into two separate herds in phase 1 but combined them in phase 2 because of grass conditions. We treated these herds as one subject in the study: they received the same treatment sequence, and their carnivore presence values are summed during the period in which these herds were grazed separately. The herds were sampled separately when they were separated, therefore summed value of carnivore presence is divided by the summed sampling pressure.

## Data collection

*Trail camera data:*  We used 34 unbaited trail cameras to detect LC presence around herds. We monitored each herd with three cameras. To place cameras, we used ArcGIS 10.7 to create a grid of 40-acre (0.18 km$^2$) cells within each grazed pasture and selected three cells at random. Within each of these three cells, we placed cameras in locations deemed likely to be visited by LCs (*e.g.*, cut lines, cow trails, stream banks, etc.). We moved cameras from pasture to pasture in response to herd movements. Each pasture overlapped an average of 17 grid cells. Therefore, a selection of three grid cells only covered about 17% of each field. However, we augmented this coverage by also conducting predator surveys within pastures; these covered the entire field and included each of the three grid cells with a camera (see section on indirect sign surveys).

We recorded all individual LC visits and numbers of visits by camera-days (*i.e.,* one day per functioning camera per field). We define Individual LC visits as visits to a camera by a single individual of a species. A new visit by an individual of the same species began when the LC was recorded on a camera 1+ hours after the last recorded photo of an LC of that species. Our criterion of one hour between photos served as an index of frequency to estimate LC presence near subject herds. Because we were not concerned with actual abundance but rather presence of LCs around cattle, we did not identify specific individuals within each LC species of interest. Therefore, we did not determine whether individual LC's were returning. We simply analyzed the presence of LCs.
*Indirect sign surveys:* The lead author (NXL) was present the majority of the study period (except for the last two weeks of October due to adverse conditions from insurmountable snowfall). NXL completed weekly surveys of LC presence focused on detecting wolf, bear, cougar and coyote activity in pastures with subject herds. These surveys included visits to each of the three trail cameras where NXL conducted systematic 100m transects along the closest animal trail to the camera. Along these transects NXL looked for scat, tracks and other signs. During each visit to a pasture, NXL also examined roads, vehicle trails, creek crossings and barbed wire fence-lines for LC signs. NXL recorded these data as presence or absence of LC by species. We examined tracks and scats whenever possible and identified them using (*Halfpenny, 1986*). Therefore, to account for uncertainty, we identified wolf and cougar presence solely from track and scat. For bears, we identified generic bear presence from hairs and signs (bear hair, scat, rubbing, and digging signs are distinctive from other LCs) but used tracks to differentiate black from grizzly bears (*Halfpenny, 1986*). Black bear tracks are smaller, with shorter nails, and the front toes form a clear arc. Grizzly bear tracks have longer nails, and the front toes are straight (*Halfpenny, 1986*). Therefore, we analyzed three sets of response variables: LC presence data derived from recorded camera visits (see more below), LC presence data from sign survey (Table 1), and the sum of the two datasets.

To reduce possible uncertainty in LC survey data, *i.e.,* from possible misidentification and non-detection of sign, the LC survey data included only NXL's observations and not those of range riders and the ranch manager. This approach also maintained the blinding of range riders to our response variable, carnivore presence. NXL blinded the second author (AT) to treatment condition during analyses. Furthermore, we cross-referenced camera photos with sign data. Not all indirect sign was captured in photos. Other aspects of indirect sign increased our confidence in species identification, such as location or type of sign.

*Analysis:* We measured the response variables of the number of records of each LC species by a simple sum of the standardized value of the number of visits per day, recorded through photos and sign surveys (binary presence or absence per survey). Data from photos were recorded as number of individual visits per camera trap day, and data from indirect sign surveys were recorded as the presence or absence of predators per sign survey. These two variables were standardized individually and then summed to determine the total presence of each LC species per day (Table 1). Also, we combined all LC presence data into one response variable (pooled LC). We tested normality of the response variables using Shapiro–Wilk tests. We found bear, coyote, and pooled LC presence data to be normally distributed. Therefore, for bears ($W = 0.94$, $p = 0.40$), coyotes ($W = 0.97$, $p = 0.86$), and pooled LC ($W = 0.95$, $p = 0.52$), we used paired test adjusted by Hills and Armitage (*Hills & Armitage, 1979*) to analyze the effect of the treatment effect and order effects. The numbers of wolf ($W = 0.84$, $p = 0.010$) and cougar ($W = 0.49$, $p = 0.00000019$) records were not normally distributed, therefore for these species we use a non-parametric Wilcoxon sum rank test to evaluate the treatment effect and order effects on wolf and cougar data (*Diaz-Uriarte, 2002*). We use Hills-Armitage paired t-tests to test the order effect of the phases, *i.e.,* whether having a treatment to pseudo-control (T-PC) sequence results in differing LC

presence relative to pseudo-control to treatment (PC-T) sequence (*Diaz-Uriarte, 2002*). Further, this method allowed us to infer whether phase 1 and 2 were consistently different. For example, if LCs were more active in fall months (September and October of our study), then phase 2 might have had greater LC presence, regardless of a herd's status as a pseudo-control or treatment.

We also used Spearman's rank tests to estimate correlation between the LC presence near herds and the frequency of range rider presence. These correlations do not provide as strong inference as the above tests of treatment effects because the daily schedule of visits by range riders was not under our control. Nevertheless, our two estimates of the change in frequency of range rider visits between phases might reveal if human supervision was associated with changes in LC presence near herds. We defined the dose effect for each herd separately, in two ways: (a) the change in the number of days in which at least one range rider was present in a herd (hereafter range rider days), and (b) the change in the number of range riders summed across each phase in a herd (hereafter dose effect). For both range rider days and dose effect, we calculated the numbers in phase 2 and subtracted the numbers in phase 1 for this within-subject analysis. We correlated both (a) and (b) to the change in presence of bears, wolves, coyotes, and pooled LCs within herds as phase 2 presence − phase 1 presence. To do so, we summed LC presence across days in each phase for each herd separately. We did not analyze cougars separately because of small sample size.

## RESULTS

We studied 2,469 adult cattle and 1,928 calves split into eight subject herds on 22,500 acres (91 km$^2$) of public grazing land between July 1 and October 31, 2020 (Table 1). No predation on cattle occurred during the experimental phases in either pseudo-control or treatment herds using a within-subjects test made possible by the crossover design. There was one confirmed livestock attack by a grizzly bear on a calf. This attack occurred during the wash-out period in a herd that was transitioning from pseudo-control to treatment, therefore it is not counted for either condition when we test for treatment effect. This attack was included as one day of grizzly bear presence in the affected herd, for the correlation between the change in grizzly bear presence and the changes in range rider days and dose effects. Eight cows died from ingesting poisonous plants, all during phase 1, and four cows died from other non-predator causes, one occurred during phase 1 and three during phase 2.

*LC presence:* During our study, we observed every carnivore species within pastures where subject herds were grazed, using cameras or indirect sign surveys for scat, track, hair, etc., regardless of treatment condition or study phase (Table 1; Figs. 3 and 4). We infer every subject herd faced some risk from predators.

*Range rider presence:* Treated herds experienced on average 2.75 times more human presence than pseudo-control herds (average 15.5 combined visits by range riders per herd *vs.* average 5.62 visits per herd by the ranch manager alone).

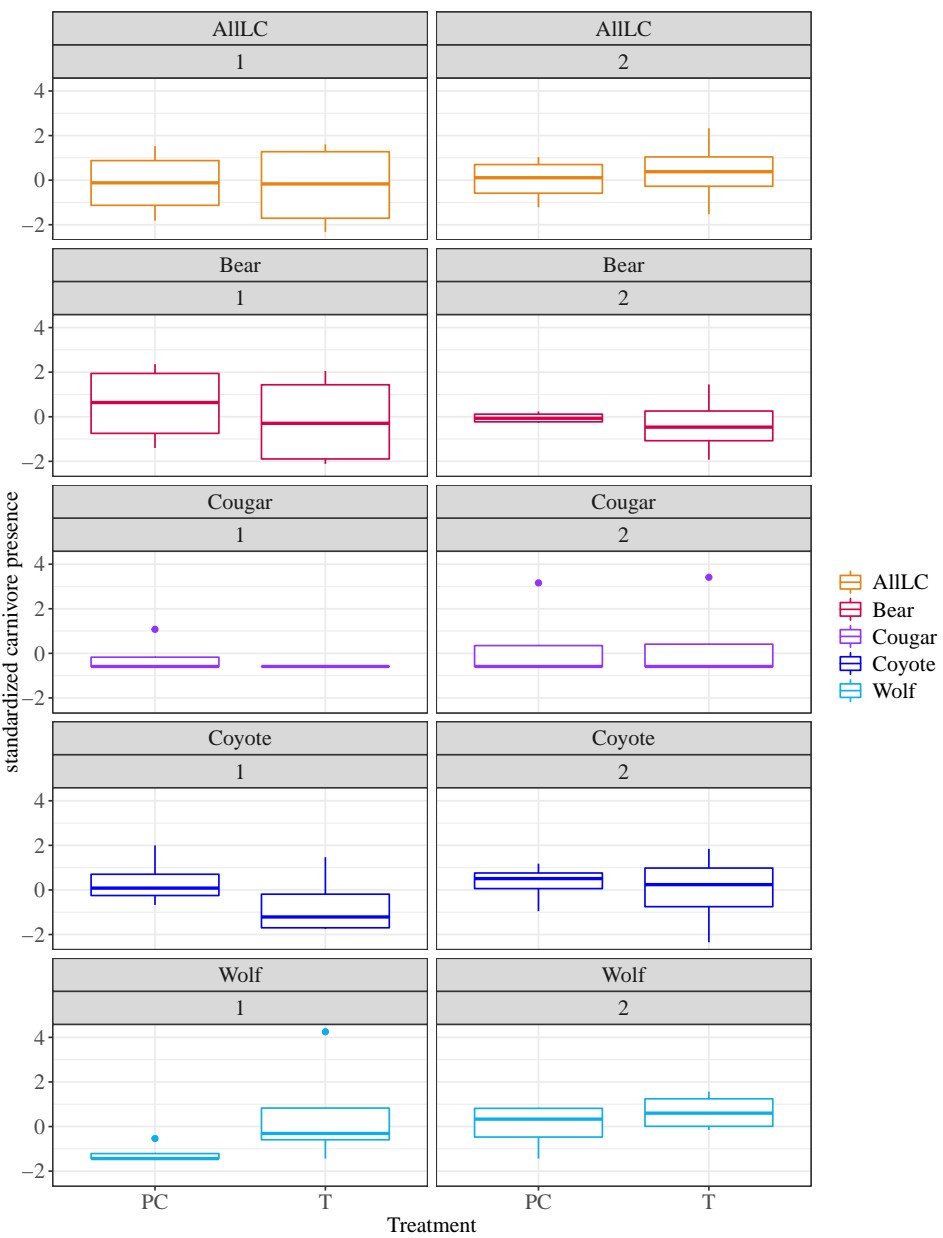

**Figure 3 Standardized large carnivore presence by treatment condition, phase and species.** Large carnivore (LC) presence standardized to include trail camera and sign survey data by carnivore species, phase (1 or 2) and treatment condition (PC, pseudo-control; T, treatment). All LC i.e. Pooled LC represents the summed presence of all species.

*Treatment effect:* The treatment conditions (treatment *vs* pseudo-control) did not predict pooled LC presence near herds in a within-subjects analysis that met the assumptions of normality and equal variance (Fig. 3, *t*-test $t(3) = -1.53$, $p = 0.89$). Therefore, we find no support for the first hypothesis that the number of range riders had an effect. Likewise, because treatments differed from pseudo-control in the experience of the added range

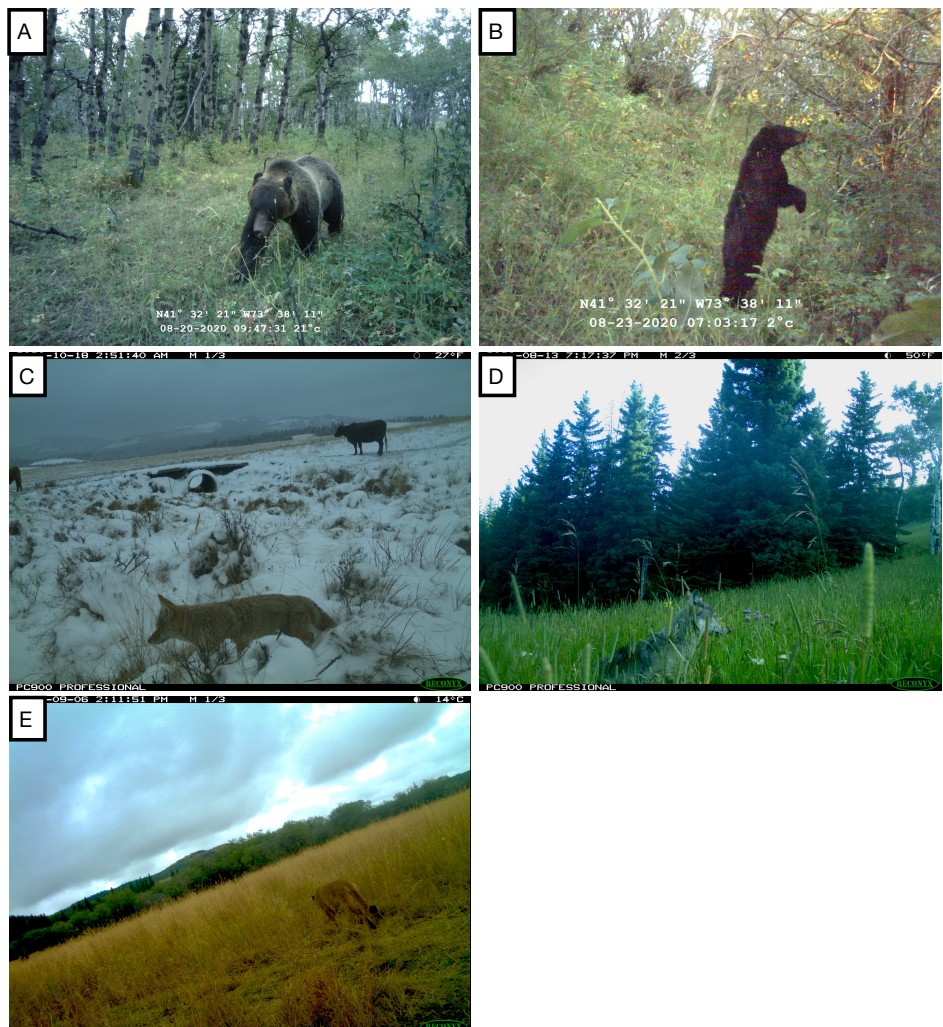

**Figure 4** **Images of predators captured by trail cameras positioned within pastures with study herds.**
(A) Grizzly bear *(Ursus arctos)* (B) Black bear *(Ursus americanus)* (C) Coyote *(Canis latrans)* (D) Wolf
*(Canis lupus)* (E) Cougar *(Puma concolor)*.

riders, we find no support for an effect of experience or inexperience of range riders.
Furthermore, the lack of treatment effect undermines the suggestion that LCs generally
were deterred from one type of range riding to the other.

*Effect of treatment on separate LC species or genus:* Bear ($t(3) = -0.21$, $p = 0.85$) and
coyote ($t(3) = -0.35$, $p = 0.75$) presence did not differ across treatment condition (Fig.
3). Cougar and wolf presence data required a Wilcoxon two tailed sum rank test ($V = 1$,
$p = 0.42$ and $V = 1$, $p = 0.25$ respectively). Therefore, species-specific presence data
conform to that for pooled LC showing no treatment effect (Fig. 3).

Period effects might confound comparisons by altering conditions across all replicates
in unison. Pooled LC presence did not significantly differ across phases ($t(3) = -0.22$,

$p = 0.84$). We observed no difference across phases for wolves ($V = 7, p = 0.625$), bears (both species, $t(3) = -0.50, p = 0.65$), coyotes ($t(3) = -0.37, p = 0.74$) and cougars ($V = 2, p = 0.789$).

Carryover effects might arise if the response to the treatment lasted after the herd was no longer being treated. While we cannot rule out the potential for carryover effects because we observed no significant difference between the treatment conditions, we attempted to eliminate any such effect by implementing a 7-day 'washout' period.

*Correlation of LC presence to range rider days and dose effects:* To account for the frequency of range rider presence on change in LC presence we examined the change in 'dosage' of range riders within a herd regardless of phase or treatment condition. We include the washout period (Table S2) (*e.g.*, three range riders could have visited a herd in the same day). Using a spearman's rank correlation we found that grizzly bear presence was weakly negatively correlated with days of range rider presence ($p = 0.066$, rho $= -0.67$), but not with dose effect, which accounted for the number of riders present summed across days in each herd ($p = 0.19$, rho $= -0.51$). There was no effect of range rider days or dosage on black bears, wolves or coyotes, nor on pooled LC. Therefore, we infer range riders were effective in protecting cattle, given losses in prior years and other Alberta ranches (see Methods). These data support the hypothesis that L-SLH reduced vulnerability but is not definitive given our pseudo-control and lack of control over the dose of range rider presence in each herd.

## DISCUSSION

We report a randomized, controlled experimental trial with crossover design to evaluate range riders practicing low stress livestock handling, L-SLH. Our experiment shows that one can graze thousands of cattle safely on vast public lands hosting grizzly bears and wolves when owners use L-SLH as a method to reduce the vulnerability of domestic animals. We conclude that L-SLH joins a growing number of non-lethal, carnivore control methods proven effective by gold standard experiments.

We find no support for the hypothesis that the risk to cattle from wolves, coyotes, black bears and cougars is affected by the single variables of number of range riders and their experience in practicing L-SLH. However, grizzly bears appeared to avoid herds exposed to regular range rider presence. Further, because L-SLH was being practiced at different dosages (one experienced rider practicing L-SLH *vs.* three riders practicing L-SLH) on every herd, and no livestock losses occurred during any treatment condition, we can conclude that L-SLH does not increase the risk of attack by LCs on cattle. We cannot rule out that a single experienced range rider practicing L-SLH every 9 days was as effective as one such experienced range rider supplemented by 1-2 inexperienced range riders, all practicing L-SLH and visiting herds every 3 days.

In our study, experienced range riders that had little or no L-SLH experience were trained quickly by an L-SLH experienced range rider. We hypothesize that functionally effective L-SLH can be trained in a short period. L-SLH is a form of livestock handling designed to reduce livestock stress, thereby increasing livestock health and yield. It has been identified

by some livestock owners in Western North America as a useful means of retraining herding behavior and reducing livestock vulnerability to carnivore attacks, especially in free-ranging herds (*Barnes, 2015*; *Zaranek, 2016*). However, this method is still relatively uncommon, many owners instead preferring handling methods that prioritize speed and efficiency over animal welfare and stress (*Hibbard, 2012*). Antithetically, when speed is prioritized, animal handling appears more difficult as animals resist herding, leading to increased stress, reduced body condition and yield (*Hibbard, 2012*; *Grandin, 1989*). This combination of factors could make free-ranging livestock particularly vulnerable to attack by carnivores (*Zaranek, 2016*; *Mech & Peterson, 2010*). Given much livestock grazing worldwide is free-ranging, L-SLH may have wide utility and co-benefits as a mobile deterrence method (*Khorozyan et al., 2020*; *Radford et al., 2020*). Range riding comes in many forms, and must be clearly defined and experimentally tested in each new application, if it is to be confirmed as effective (*Parks & Messmer, 2016*). A lack of consensus on its definition and methods of use reduces its functional and perceived effectiveness, and risks wasting government and community resources (*Parks & Messmer, 2016*). We propose that further research is needed to compare different forms of animal handling to determine whether L-SLH reduces vulnerability of livestock when compared to 'traditional' handling methods. Furthermore, it should be compared in more species of livestock, though there is some evidence that it may be effective with sheep (*Stone et al., 2017*), and L-SLH can be used on animals from cattle to chickens (*Hibbard, 2012*).

## Co-benefits

An assumed benefit of range riding is the ability of range riders to quickly observe and manage all sorts of problems affecting the herd. For example, during our study there were eight recorded cattle mortalities from cattle ingesting toxic plants, such as larkspur, water hemlock or saskatoon blooms (*Majak, Brooke & Ogilvie, 2008*), and four mortalities from other non-predator causes. This is not uncommon in our study area according to the ranch manager, and occurred more regularly earlier in the grazing season, during the first phase of our study, when the majority of poisonous plants are at their most toxic (*Majak, Brooke & Ogilvie, 2008*). Range riders or the ranch manager identified these areas and were either able to remove dead stock or increase their own presence within these herds.

A primary attractant of grizzly bears in this region of Alberta are dead animals (*Morehouse, 2016*), so the presence of poison killed cattle might be attracting bears into the study area. This may explain why our results show a non-significant increase in presence of both species of bears during phase 1 of the study (Fig. 3). Attacks by bears on livestock did not increase however, either due to the supplemental feeding provided by the already dead animals, or the increased presence of range riders during these times. We did not observe that these dead animals attracted wolves or other carnivores. *Morehouse & Boyce (2017)* found that diversion feeding of grizzlies in Alberta resulted in a few dominant males protecting the food source against other individuals. Therefore, depending on the individuals attracted to the dead livestock, bears could have repelled other bears or species from the area. Evidence is mixed regarding whether diversionary feeding of carnivores is effective (*Garshelis et al., 2017*) at reducing conflict (*Morehouse & Boyce, 2017*; *Steyaert*

*et al., 2014*), attracts bears and could increase conflict (*Kavčič et al., 2015*) or is effective by diverting carnivores to other sources (*Stringham & Bryant, 2015*). It is difficult to disentangle the effects of human presence, diversion feeding, and competition between bears and other carnivores. We did observe a large negative correlation between the numbers of days with range riders and the presence of grizzly bears, with fewer bears present when range riders increased their presence. However, there was no dose effect of the number of range riders within a herd summed over the study period. In other words, grizzlies might have responded to presence or absence of range riders in a day, but not to the number of individual riders present throughout the study phase. Presumably the direction of causality would not be the reverse, given range riders were employed to deter predators.

Further research is needed to determine the potential for other co-benefits of L-SLH. For example, reduced numbers of livestock losses, either by avoiding poisonous plants or reducing carnivore attacks would increase annual yield for ranchers. However, reduced stress could also significantly increase livestock yields, and therefore improve financial returns at the end of the season (*Stringham & Bryant, 2015*). Such experiments have never been conducted in conjunction with L-SLH.

## Attacks on livestock

The presence of carnivores throughout the study period suggests that risk existed. Furthermore, shortly before the study began, a wolf attack was confirmed within one of the study area's cattle herds. Wolf and grizzly bear attacks on calves were commonly reported throughout the ecosystem in which we worked in the past, with 92 livestock losses reported in the previous 5 years (*Alberta Environment and Parks, 2020*). While we continued to observe carnivores during our experiment, we received no news of heightened attacks. Carnivores did not on average change their frequency of presence within either treatment or pseudo-control herds (but see discussion of wolves and coyotes below). No carnivores attacked livestock during either treatment or pseudo-control condition (but see below for an attack during the wash-out period). Despite a consistent presence of grizzlies, black bears, wolves, cougars and coyotes throughout our study period, individual predators or their foraging groups did not switch to pseudo-control or treatment herds when confronted with both. Therefore, we find no support for the hypothesis that large carnivores moved from the better-protected herds to less well-protected herds within a large public grazing land allotment. Therefore, our results seem contrary to claims that non-lethal methods simply displace predators to less-protected livestock.

Indeed, such spill-over effects have been reported for lethal control. Several studies found lethal control counter-productive (*Van Eeden et al., 2018*; *Treves et al., 2019*). *Santiago-Avila, Cornman & Treves (2018)* estimated that Michigan farms treated with lethal methods experienced a reduction in the future risk of livestock attacks by approximately 25%, while at the same time, untreated neighboring farms experienced an increase in future risk of approximately 75%. *Grente et al. (2021)* found a similar failure to attain desired outcomes of lethal control in a majority of nine study areas in France.

During our study's wash-out period, grizzly bear(s) apparently killed a calf in the week preceding when we would switch that herd from pseudo-control to treatment. During the wash-out, all eight herds experienced the lowest frequency of range rider supervision because only the ranch manager visited each herd. Furthermore, the dead calf's herd experienced an average amount of grizzly bear presence in phase 1 (four grizzlies, mean of all herds = $4.75 \pm 4$), it had a lower-than-average number of days with range riders (4 days with range riders, mean of all herds = $10.5 \pm 6.7$) during this phase. This suggests that the lack of human presence provided an opportunity for attack but did not attract grizzlies. While this single attack does not change the statistical significance of the treatment effect, it does suggest that the addition of two newly trained range riders is more effective at reducing risk to cattle than the ranch manager working alone because it increased the number of days riders could be in the herd. Therefore, it is likely that there is a dose effect of L-SLH range riders, but more research is needed to confirm and evaluate how much time should be spent with a herd.

## Wolves

Though the effect was not significant, wolves were observed slightly more frequently during phase 2 and near treatment herds during phase 1 of the study (Fig. 3). The increased presence of wolves later in the season may result from wolf breeding behavior. Wolves select breeding sites further from human activity where humans persecute them, as they do on this landscape. This may explain why fewer wolf signs were observed during phase 1, which occurred while pups are young and wolf packs remain close to breeding sites (*Mech & Peterson, 2010*; *Sazatornil et al., 2016*). Phase 2 occurred during the autumn (September-October), when pups are older and wolf packs become nomadic, increasing the likelihood that we would observe them in our study area (*Wilson, Bradley & Neudecker, 2017*; *Neufeld, 2006*; *Packard, 2010*). The non-significant increase in the presence of wolves in treatment herds during phase 1 compared to pseudo-control herds may reflect a larger number of calves in the treatment condition during phase 1, or the novelty of the new range riders. First, calves are smaller and weaker than mature cattle, and therefore are more vulnerable to predation. Smaller calves are also less independent and rely on their mothers and the herd for protection (*Flörcke & Grandin, 2013*). However, if wolves were attracted to the increased number of calves in the treatment condition during phase 1, it is probable that range riders aided in reducing the vulnerability of these herds, leading to no attacks on the calves. We did not observe this same trend in phase 2 when the pseudo-control condition contained more calves. Wolves in Alberta may increase predation on livestock in late summer and fall (during phase 2) (*Dorrance, 1982*), and appear to preferentially select for cattle less than 9 months old. Therefore, we would expect to observe more wolves in pseudo-control herds in phase 2, where there were more calves. During this time calves are larger in size, but they are also more independent (*Reinhardt & Reinhardt, 1981*), which may increase their vulnerability in different ways. For example, they wander further from their mothers as they begin to wean off milk and their mothers become less vigilant over them (*Flörcke & Grandin, 2013*; *Reinhardt & Reinhardt, 1981*). However, there was no

difference in wolf presence between treatment conditions during phase 2, consistent with the novelty of new range riders wearing off.

The second possibility for increased wolf activity in treatment herds in phase 1 is the novelty of new range riders. We presume the intelligence of wolves leads them to investigate novelties such as range riders (*Range & Virányi, 2013*). Carnivores explore novel situations to gain information about their environments and territories (*Much et al., 2018*). Repeated exposure to certain circumstances can, however, reduce curiosity and exploratory behavior (*Mettke-Hofmann et al., 2006*). Many individual wolves avoid new objects and circumstances (*Mettke-Hofmann et al., 2006*). This avoidance is thought to be a primary driver of the success of fladry, a form of fencing that uses evenly spaced flagging, to reduce wolf encroachment into fladry surrounded areas (*Eklund et al., 2017*; *Davidson-Nelson & Gehring, 2010*; *Musiani et al., 2003*; *Musiani & Visalberghi, 2001*). However, despite fladry deterrent effects in keeping wolves out, in most studies which have recorded wolf approaches to fladry, more wolf approaches were recorded in proximity to fladry fencing, than to control areas where no fladry was installed, despite wolves rarely if ever crossing the fladry barriers (*Davidson-Nelson & Gehring, 2010*; *Musiani et al., 2003*). Therefore, our finding of increased number of wolf visits in treatment herds may have resulted from the novelty of the new range riders. If this conjecture is correct, an increased number of observations of wolves does not imply increased risk to livestock, but instead an opportunity for wolves to learn about their environment. This learning is an important aspect of the deterrence work of range riders (*Much et al., 2018*). If wolves explore their territory and learn that range riders are a threat or that livestock are not vulnerable, we might expect them to become accustomed to the presence of range riders and focus their energy on hunting wild prey. This may explain why, despite more frequent visits by wolves in phase 2 of our study, there was no increase risk for cattle between the two study phases.

## CONCLUSION

We observe that when properly executed L-SLH protects cattle with fewer riders. Furthermore, it is a method that is quickly taught. Our recommendations are that (1) L-SLH be tested in a randomized controlled experiment against non-L-SLH (*i.e.,* 'traditional') livestock handling and/or a true control. This would help to determine whether other forms of human presence deter, attract or have no effect on predator attacks on livestock. True controls might involve a saddled, riderless horse with the cattle herd or visits by humans who do no herding at all. However, we demonstrate in this experiment, how difficult it is to implement a true control, as many livestock owners are unwilling to leave their herds unattended; (2) L-SLH methods should be studied to examine the number of newly trained range riders that are optimal for predator deterrence and cost effectiveness. The results of this study suggest that fewer riders may be just as effective on the predators being deterred, but that some predators, such as grizzly bears, may require a minimum amount of range riders practicing L-SLH. Further, studies comparing numbers of newly trained riders would produce evidence regarding what level of training must be attained for effective predator deterrence, particularly as there are few L-SLH-veteran range riders

working today. Therefore, given a lack of further research on range riding, and a lack of consensus on the efficacy of other forms of range riding, L-SLH should be prioritized as the only form of range riding, to our knowledge, to have been experimentally tested.

Methods to reduce risk of attacks on livestock present many benefits, particularly as free-ranging livestock occur throughout the world and present a challenge to co-existence with carnivores. Human-caused mortality, often in response to perceived conflicts, is the primary form of mortality in large carnivore populations, and risks undermining conservation efforts to restore carnivore populations (*Ripple et al., 2014*). By not harming carnivores through displacement or unbalancing social structures (*e.g.*, wolf packs), L-SLH also presents itself as a non-lethal predator control method which is effective for both target and adjacent properties without harming the carnivores (*Haber, 1996*).

## ACKNOWLEDGEMENTS

We would like to thank Joe Engelhart and the members of the Spruce Ranching Co-op for being our research partners and allowing us to design this experiment with their livestock. We would like to thank our range riders for their hard work and dedication to this project.

### Funding

This work was supported by the National Geographic Society (NGS-67956-20), the Natural Sciences and Engineering Research Council of Canada (PGS-D3-545968 - 2020), the Animal Welfare Institute (Christine Stevens Wildlife Award 2019) and the Yellowstone to Yukon Conservation Initiative (Partner Grant 2019). The funders had no role in study design, data collection and analysis, decision to publish, or preparation of the manuscript.

### Grant Disclosures

The following grant information was disclosed by the authors:
National Geographic Society: NGS-67956-20.
Natural Sciences and Engineering Research Council of Canada: PGS-D3-545968 - 2020.
Animal Welfare Institute (Christine Stevens Wildlife Award 2019).
Yellowstone to Yukon Conservation Initiative: 2019.

### Competing Interests

The authors declare there are no competing interests.

### Author Contributions

- Naomi X. Louchouarn conceived and designed the experiments, performed the experiments, analyzed the data, prepared figures and/or tables, authored or reviewed drafts of the article, and approved the final draft.
- Adrian Treves conceived and designed the experiments, analyzed the data, authored or reviewed drafts of the article, and approved the final draft.
## Animal Ethics

The following information was supplied relating to ethical approvals (i.e., approving body and any reference numbers):

Institutional Animal Care and Use Committee at the University of Wisconsin-Madison provided a study exemption letter (attached).

## Field Study Permissions

The following information was supplied relating to field study approvals (i.e., approving body and any reference numbers):

The Government of Alberta does not require field permits because we did not collect any samples, nor did we use any invasive observation methods.

## Data Availability

The raw data and code are available in the Supplementary File.

## Supplemental Information

Supplemental information for this article can be found online at http://dx.doi.org/10.7717/peerj.14788#supplemental-information.

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
