# Peer review of "Low-stress livestock handling protects cattle in a five-predator habitat"

_PeerJ, doi:10.7717/peerj.14788_

## Round 0.1 · original submission · Major Revisions

· Academic Editor

Major Revisions

Both reviewers have suggested major revisions, with many important comments which need to be addressed. Hence my recommendation to revise the manuscript paying close attention to all their comments.

Reviewer 1 ·

Basic reporting

This study explores the effectiveness of horse riders accompanying grazing cattle in reducing cattle losses to five species of large carnivores. Although the paper is solid, I do not see valuable results which can be published, what is very pity. Therefore, I leave it to the editor’s discretion whether this manuscript should be finally suggested for a major revision or rejected. More details are given below

Experimental design

This study is very interesting methodologically and practically for future replications and improvements. The authors applied a strong study design, performed appropriate data analysis, and wrote a good-quality paper. I like how the authors borrowed the crossover experimental technique from medicine and applied it to practical conservation.

Validity of the findings

No cattle were killed in random treatment (ranch manager + 1-2 range riders) and control (ranch manager alone) samples, therefore the effectiveness of cattle herding could be neither supported nor refuted. Only one calf was killed by a bear during the 7-day washout period between the treatment and control periods. As the authors mentioned, the difference between treatment and control samples could be diluted due to the training provided by the ranch manager to range riders. I think that this dilution also could result from the use of deterrents in both treatment and control herds to keep carnivores away. Moreover, absence of cattle losses could result from a very short duration of the study (123 days over 4 months) – as carnivore attacks on livestock are generally random in space and time, such studies should be much longer to collect a sufficient sample of losses.

However, on L341-342 the authors conclude that riders were effective in reducing cattle losses compared to higher losses in previous years. This result is not supported as no data from previous years were used in this study. Therefore, I suggest the authors to re-analyze their data employing a different approach, Before-After-Control-Impact (BACI), which accounts for differences in cattle losses before and after treatment in treatment and control samples. As the authors write in Methods, “human supervision is scarce” so using placebo-control and not pseudo-control is possible. But in this case cattle attendance vs. non-attendance should be carefully recorded for time (before-after) and study sites (treatment-control).

Additional comments

Other comments:

L141 – what is the reason of under-reporting cattle losses? I would expect over-reporting to be more logical, e.g. to justify carnivore killing in retaliation or to demand compensations.
L176-178 – so the riders worked as a pair in bear areas but alone in non-bear areas? It is not described when the riders worked alone. In other words, treatment samples included the range manager with one rider in non-bear areas and two riders in bear areas, right? This should be explained a bit more clearly.
L224-229 – how many grid cells covered each pasture, on average? Just to understand the proportion of three camera-trapped grid cells to other non-trapped cells. Non-detection of carnivores by few camera traps can be moderate to high, especially if carnivores do not penetrate deep into the pasture. Could the authors write a bit more on this?
L236 – what does it mean “we did not identify individual LCs”? Later the authors write that they estimated LC presence from camera traps.
L262-264 – the description of the response variable is not clear. Did you sum up the presence/absence data or the numbers of records per day? Also, what do you mean “standardized”? For example, during three days you recorded 4 camera/sign records on the first day, none on the second day and 3 on the third day. Did you measure your response variable as 7 (4, 0, 3) or 2 (1, 0, 1)? Please explain it here and above in the text where you described three sets of the response variable (L249-250). On L265, you write that you tested your response variables for normality, so I think you summed up the numbers of records. Then, please rename the response variable from presence/absence (which is binary by default) to the number of records.

Minor comments in Methods:

L123 – change acres to m2 or km2 here and elsewhere, wherever present, or indicate in the parentheses after acres.
L146 – write “Introduction” instead of “intro”
L149 – change to “consideration of”
L171 – better to replace “bleeding” by “blurring”
L175 and elsewhere – when in plural, “riders” should begin with a lower-case letter
L178 – change to “helped train”
L182-183 – change to “as described on Fig 1”
L185 – change to “original”
L200-201 – change “were” to “was” and “is” to “was”. Delete “following” and place the reference 25 immediately after “subject”
L213 – change to “minimized”
L214 – change to “avoided”
L215-216 – this sentence is confusing. 16 trials is in total, for 8 treatments and 8 controls of 8 herds. So better to re-word to something like “We studied 8 herds (replicates) and therefore had 16 trials in total, including 8 treatments and 8 controls”
L230 – change to “numbers”
L231 – change to “defined individual LC”
L235 – change “are” to “were”
L247 – change to “from”
L251 – change to “e.g., from possible misidentification and non-detection of”
L249-250 – change to “Therefore, we analyzed three sets of the response variable for each LC species: presence data derived from cameras, presence data from sign surveys (Table 1), and the sum of these two datasets”
L252-253 – change to “only NXL’s observations and not those of range riders and the ranch manager”
L253-254 – this sentence is redundant and should be deleted
L256 – change to “blinded the second author (AT)”
L258 – change to “increased”
L258-261 – these sentences reduced your confidence in species identification and not increased it, as you could not differentiate black and brown bears from DNA. These sentences should be deleted
L262 – change “estimated” to “measured”
L266 – change to “Shapiro-Wilk test” and to “and pooled LC”. Add the statistics of normality in the parentheses
L267-268 – change to “we used paired test adjusted by Hill and Armitage (1979) to estimate time-mediated treatment effects in crossover experiments”. Please give a reference to Hill and Armitage (1979), it comes from medicine and not known in wildlife sciences and conservation
L268 and elsewhere – “presence” should be replaced by the number of records, so here it would be “The number of wolf and cougar records…”. Otherwise, it sounds incorrect – presence is not normal. Presence/absence is never normal, it is binary.
L269 – give the normality statistics in the parentheses. Change to “we used non-parametric”
L270 – on each what? Change to “We used Hills-Armitage”. Also, does Hill-Armitage t-test work on non-normally distributed samples? Above you indicated it only for normal samples. Please explain
L273 – change to “allowed”
L277-278 – change to “rank test to estimate correlation between the LC presence near herds and the frequency of range rider presence”
L281 – change to “if human supervision”
L287 – change to “both (a) and (b) to the change”
L289-290 – this part is intuitively clear: “assigning a 1 for any sign or photo of that LC species and zero if no sign or photo had been recorded”. If we sum up something, we assign 1 for a unit and 0 for nothing. This part should be deleted
L291 – change to “because of low”

Reviewer 2 ·

Basic reporting

The study carried out by Shima et al. titled “Low-stress livestock handling protects cattle in a five-predator habitat”. They setup for controlled experiment to evaluate low-stress livestock handling (L-SLH) and in my opinion study seems significant to mitigate the livestock depredation by large carnivores in the area. Experiment was set using a rancher (two trained but new [with limited experience] and one was experienced which act as a pseudo-control, if I correctly understand the experimental set-up). The theory behind this is to increase the presence of the human as a form of ranchers to deter the large carnivore or which increase the stress on predator. But how it helps to cattle to stay/move as a herd? Not clear. Weather this method will create the stress on livestock/ or predator is not clear. It is difficult to understand what results suggest, L-SLH practices are useful or not with current description not. At some extent the results and statistical analysis does not support it. Further, methodology is not clearly defined and confusing like what was control experiment [need more explanation] and what expected results it leads to. For examples what is pseudo control. How the ranchers will create the stress, why it was set. The cross reference is given but it should be clearly understandable in this draft for clarity. Further, this experiment only conducted with cattle which limits the study, does this study has implication in other type of livestock. It is obvious that, human presence will deter the presence of the carnivores, then what is new in it, will it ultimately help to reduce the man power to reduce the cost. Further, the experiment also includes the wolf, is wolf capable to prey upon the cattle? And how frequently. Further, how many total cameras were placed, how frequently LC were captured and what was abundance the, required more details. Further, livestock vulnerability to carnivore attacks is reduced by L-SLH or presence of humans it need to clearly explained in through details in methods.
Minor comments
L-225: Only three cameras were sufficient to capture animal presence around the herd?
L242-how the sign of different species was differentiated.
What is difference between treatment-pseudo-control (T-PC) and pseudo-control-treatment (PC-T)?

L370: If your results do not support, then how L-SLH may have wide utility and co-benefits?
How it is considered as Low stress.
L481-490: Authors, discussed that, wolf presence increase but did not attacked the animal due the presence of the range rider which indicates, to keep wolf away from the herd? But what if there is no human presence, or if the wolfs are really capable to hunt cattle (especially adult one), how many past incidences recorded of wolf depredation on the cattle from the areas? Need to be discussed here.
I would be able to fully comments on the paper on ground of above suggested comments and would happy to see revised version.

Experimental design

Experimental design is standard but need to provide more details

Validity of the findings

Though the study was conducted two phases and thus results are validated, some of details are lacking which mentioned in main report.

Additional comments

NA

---

## Round 0.2 · Minor Revisions

· Academic Editor

Minor Revisions

The authors have incorporated the comments and suggestions suggested by reviewers but minor revisions are still needed.

Reviewer 1 ·

Basic reporting

The authors did a good job to comprehensively address the comments made by both reviewers. I just suggest to make the following minor technical corrections before the paper can be accepted.

L88 – change to “animal’s”
L89 – change to “handler’s”
L90 – change to “As ungulates”
L187 – change “Riders” to “riders”
L208 – change “Riders” to “riders”
L256 – change “Individual” to “individual”
L257 – change to “the presence”
L272 – change to “(see more below), LC presence data”
L277 – change to “the second author”
L284 – change to “the number”
L285 – change to “the presence”
L292 – write exact p values instead of p<0.05, change to “The numbers”
L302 – change to “and the frequency”
L304 – change to “small sample size”
L380 – change to “practiced”
L383 – change to “the risk”
L406-407 – change to “cattle and chickens”, without the first capital letters

Experimental design

Suitable and well-described

Validity of the findings

Strong enough

Additional comments

N/A

Reviewer 2 ·

Basic reporting

Authors have addressed all comments except minor clarification.
Comment 12: L242-how the sign of different species was differentiated.
Is it effective to identify the brown bear and black bear scat through morphology, without genetics and how tracks were used to identify the species? is not clear.

Experimental design

Methods described with standard and statistically supported.

Validity of the findings

Finding are well tested in field.

---

## Round 0.3 · accepted · Accept

· Academic Editor

Accept

The reviewer makes a positive response to your manuscript hence my recommendation is to accept.

Reviewer 1 ·

Basic reporting

I have no more comments and suggest to accept the revised version as it is.

Experimental design

Strong and appropriate.

Validity of the findings

Strong.